# Trash to Treasure: How Insect Protein and Waste Containers Can Improve the Environmental Footprint of Mosquito Egg Releases

**DOI:** 10.3390/pathogens11030373

**Published:** 2022-03-18

**Authors:** Megan J. Allman, Aidan J. Slack, Nigel P. Abello, Ya-Hsun Lin, Scott L. O’Neill, Andrea J. Robinson, Heather A. Flores, D. Albert Joubert

**Affiliations:** 1Institute of Vector-Borne Disease, Monash University, Melbourne 3800, Australia; slack.aidan@gmail.com (A.J.S.); nigel.abello@gmail.com (N.P.A.); heather.flores@monash.edu (H.A.F.); 2Department of Microbiology, Monash University, Melbourne 3800, Australia; 3World Mosquito Program, Monash University, Melbourne 3800, Australia; ya-hsun.lin@worldmosquito.org (Y.-H.L.); scott.oneill@worldmosquito.org (S.L.O.); albert.joubert@worldmosquito.org (D.A.J.); 4School of Chemistry, Monash University, Melbourne 3800, Australia; andrea.robinson@monash.edu

**Keywords:** *Aedes aegypti*, mass release, egg release, diet, environment, waste, *Wolbachia*

## Abstract

Release and subsequent establishment of *Wolbachia*-infected *Aedes aegypti* in native mosquito populations has successfully reduced mosquito-borne disease incidence. While this is promising, further development is required to ensure that this method is scalable and sustainable. Egg release is a beneficial technique that requires reduced onsite resources and increases community acceptance; however, its incidental ecological impacts must be considered to ensure sustainability. In this study, we tested a more environmentally friendly mosquito rearing and release approach through the encapsulation of diet and egg mixtures and the subsequent utilization of waste containers to hatch and release mosquitoes. An ecologically friendly diet mix was specifically developed and tested for use in capsules, and we demonstrated that using either cricket or black soldier fly meal as a substitute for beef liver powder had no adverse effects on fitness or *Wolbachia* density. We further encapsulated both the egg and diet mixes and demonstrated no loss in viability. To address the potential of increased waste generation through disposable mosquito release containers, we tested reusing commonly found waste containers (aluminum and tin cans, PET, and glass bottles) as an alternative, conducting a case study in Kiribati to assess the concept’s cultural, political, and economic applicability. Our results showed that mosquito emergence and fitness was maintained with a variety of containers, including when tested in the field, compared to control containers, and that there are opportunities to implement this method in the Pacific Islands in a way that is culturally considerate and cost-effective.

## 1. Introduction

Emerging strategies targeting vector control and replacement offer promising long-term solutions to mosquito-borne epidemics. However, the large scale at which these diseases exist is a significant hurdle [1,2]. The most widespread of these are dengue, Zika, chikungunya, and yellow fever, with an estimated 390 million dengue virus (DENV) infections occurring annually, and approximately half of the global population at risk of DENV infection [3,4]. *Aedes (Ae.) aegypti* is a major vector of DENV, Zika virus (ZIKV), chikungunya virus (CHIKV), and yellow fever virus [5,6]. Alarmingly, as vector habitats have expanded, so has the incidence of dengue cases, increasing over 8-fold from 2000 to 2019 [7,8], indicating the ineffectiveness of historical techniques and the need for sustainable biocontrol methods that are scalable.

*Wolbachia* introgression offers one such solution. *Wolbachia* is an endosymbiotic bacterium that inhibits virus replication in *Ae. aegypti* when stably transinfected [9,10,11,12]. *Wolbachia*-infected *Ae. aeygpti* populations have been successfully established and maintained in the field since 2011 [13,14], and a recent randomized control trial in Yogyakarta, Indonesia, showed this method can reduce dengue incidence by 77% and hospitalizations by 86% [15].

Typically, *Wolbachia*-infected mosquitoes can be released via two pathways—one, as fully developed adults and two, by allowing eggs to hatch and develop in containers in the field. For adult releases, mosquitoes are reared at high-density in large trays and aliquoted as pupae into release tubes to emerge [16,17]. For egg releases, eggs are placed in a single-use, plastic-lined cardboard mosquito release container (MRC) in the field containing water and sufficient larval food allowing for the development of immature stages that will emerge from the container as adults. Egg releases are advantageous compared to adult releases because they can reduce onsite resource requirements, as personnel are not required to handle mosquitoes at immature aquatic stages, and can be leveraged to increase public acceptance and community engagement [14,18,19,20]. Egg releases have successfully been used by the World Mosquito Program (WMP) to establish and maintain *Wolbachia* frequency in parts of Queensland, Australia and Yogyakarta, Indonesia [14,19,20]. However, the release method could potentially lead to unwanted environmental outcomes, such as waste accumulation, due to single-use MRCs. In this study, we tested an egg release strategy using MRCs with a reduced environmental footprint through improved larval diet and MRC sourcing and management.

A current high-performance larval diet, as developed by the International Atomic Energy Agency (IAEA) [21], sources protein from beef liver powder. Insect meal offers a promising alternative protein source as insect rearing requires less resources, such as water, land, and feed, and produce approximately 1% of the greenhouse gas emissions compared to other protein sources [22,23,24,25]. In addition, insect meals provide similar protein profiles to beef liver powder and contain all essential amino acids [25,26,27]. Hence, we substituted cricket and fly meal for beef protein and compared mosquito development and adult fitness when reared on different protein sources.

Previous egg release projects utilized eggs on a substrate, a labor-intensive strategy with limited scalability potential [14]. Encapsulating eggs improves these issues and has been developed and tested with success in mass-mosquito release trials [28,29,30]. Here we developed an alternative, environmentally friendly egg encapsulation method which could increase the scale at which eggs and food can be transported on-site, allowing the powdered larval diet to be used in the field and removing plastic waste during transportation.

Egg releases can inadvertently increase waste in the deployment phase through the placement of containers into the environment. This is particularly a problem because solid waste accumulation can be associated with *Aedes*-borne diseases due to the provision of breeding sites [31,32,33]. Waste accumulation also increases the risk factor of other diseases, such as diarrhea and pneumonia, through decreased sanitation and contamination of freshwater resources [34]. The Pacific Island nation of Kiribati, where part of this study was conducted, has a significant waste issue due to its remote location. The majority of waste is shipped for offshore processing, and there is a historical lack of landfill or recycling facilities, with the first landfill being built in 2002 [35,36]. The potential negative health impacts of waste mismanagement is of growing concern within this community [37]. A novel and creative concept is the prospect of recycling commonly found waste containers—such as aluminum cans, PET bottles, tin cans, and glass bottles—as MRCs. This study used Kiribati as a case study to investigate whether these containers are viable MRC options that support mosquito development and fitness.

## 2. Materials and Methods

### 2.1. Mosquito Strains and Maintenance

Two mosquito strains were used throughout this study: Australia (Aus) *w*Mel *Ae. aegypti* and Kiribati wild-type *Ae. aegypti* (Appendix A). The Aus *w*Mel strain was generated by O’Neill et al. [20]. *Ae. aegypti* infected with the *w*Mel strain of *Wolbachia* initially developed by Walker et al. [38] were collected from field release sites in 2012 in Cairns, Australia and were outcrossed with wild-type *Ae. aegypti* sourced from Townsville for six generations [20]. Colonies were maintained under standard laboratory conditions in a climate-controlled insectary at 26.5 °C, 70% relative humidity (RH) with a 12 h:12 h light:dark cycle, as described by Ross et al. [39]. Colonies were regularly screened for *Wolbachia* by qPCR and assessed for fitness compared to *Wolbachia*-uninfected lines via assays described by Walker et al. [38] and Xi et al. [40]. The field experiment conducted in Kiribati used wild-type eggs collected from South Tarawa using ovitraps with cloth strips for oviposition.

### 2.2. Mosquito Rearing

#### 2.2.1. Larval Diet

The control beef-based diet was prepared by thoroughly grinding and mixing beef liver powder, tuna meal, and brewer’s yeast together as described by Puggioli et al. [21]. Alternative protein sources were tested by substituting beef liver powder for black solider fly meal (fly-based diet) or cricket meal (cricket-based diet). The liquid diet version was prepared by mixing solid components with reverse osmosis (RO) water to form a 7.51% slurry of each diet. Food components were stored at 4 °C.

#### 2.2.2. Four-Diet Comparison

One hundred and fifty larvae were reared in insectary cups (820 mL plastic cup) with 500 mL of RO water. Five cups per diet were prepared. A liquid diet feeding regime was followed for beef-, fly-, and cricket-based diets (Appendix A). Mosquito larvae on a Tetramin (Tetra) diet were monitored daily and fed when food supply was low.

#### 2.2.3. Test of Food and Eggs Inside Capsules

To prepare egg capsules, eggs were quantified by counting and gently brushed from the paper substrate into a water-soluble capsule using a small paintbrush. Capsules were then topped with 285 mg of food. Ten capsules for each diet combination (beef, fly, and cricket) were prepared and reared in individual cups with 300 mL of RO water.

#### 2.2.4. Waste Container Comparison

To compare mosquito emergence from recycled waste containers, capsules with eggs and cricket-based diet were prepared and hatched in five different container types: aluminum can (355 mL), PET bottle (600 mL), tin can (300 mL), glass bottle (355 mL), and plastic cup (820 mL) as control, each with 300 mL of RO water and with five replicates of each container. For each container, the hatch rate of encapsulated eggs, as well as wing lengths and *Wolbachia* density for emerged mosquitoes, were determined as described below.

### 2.3. Hatch Rate

To conduct hatch rate tests, the number of larvae in an individual container was counted 48 h after egg submersion. Larvae were returned to their corresponding containers after counting and allowed to develop to adulthood. Hatch rate was calculated as the number of larvae as a proportion of the number of eggs hatched per container.

### 2.4. Emergence Rate

Emergence rate was determined 14 days post-hatching and calculated as the number of emerged adults as a proportion of the total eggs hatched per container.

### 2.5. Wing Length

Four adult females from each replicate (total 16–20 per group) were collected from five-day old mosquitoes. Wing length was determined based on the method described by Packer and Corbet [41]. One wing was removed from each mosquito under a dissecting microscope with an eyepiece micrometer, and wing length was measured from the distal edge of the alula to the end of the radius vein, excluding fringe scales.

### 2.6. Wolbachia Density

Six-day-old adult females were collected (40 females per container group), individually placed into 96-well plates and homogenized in extraction buffer consisting of 50 μL squash buffer (10 mM Tris, pH 8.2; 1 mM EDTA; 50 mM NaCl) supplemented with 25 μg/mL proteinase K (Bioline) and a 2 mm glass bead (Pacific Laboratory Products). Samples were incubated in a thermocycler (5 min at 56 °C, 5 min at 98 °C, and held at 12 °C). Mosquito homogenates were diluted ten-fold using AE buffer (Qiagen). Total relative *w*Mel density was estimated by multiplex quantitative polymerase chain reaction (qPCR). qPCR reactions were performed in 10 μL total volume containing 5 μM 2 × LightCycler 480 Probes Master reaction mix, 2.5 μM primer, 1 μM probe *(Wolbachia surface protein [wsp]* and *Ribosomal Protein S17 [RpS17**]*), and 3 μL diluted (1:10) adult homogenate (see Appendix A for probe and primer sequences). Cycling was performed using a LightCycler 480 II system (Roche) with 1 cycle at 95 °C for 5 min, followed by 45 amplification cycles of 95 °C for 10 s, 60 °C for 15 s, and 72 °C for 1 s. To analyze qPCR data, normalized expressions (NEs) were calculated using a delta Ct method [42]: NE = 2^C^_q_^reference^/2^C^_q_^Target^, where *RpS17* is the reference gene and *wsp* is the target gene [43,44].

### 2.7. Statistical Analysis

Data analysis was undertaken using R v 1.4.1717 and visualized using GraphPad Prism v 9.2. Normality was checked using the Shapiro–Wilk test and assumptions using diagnostic plots and residual simulation plots [45]. We performed a linear model (parametric data), a Kruskal–Wallis or Mann–Whitney test (non-parametric data), or a generalized linear model (proportional data) [46,47,48]. Modelling was followed by ANOVAs to compare treatment effect [49]. In the case that significant interactions were identified, we used Tukey’s *p*-value adjustment method for pairwise comparisons [50].

### 2.8. Kiribati Case Study

#### 2.8.1. Kiribati Waste Container Field Trial

Four types of waste containers were collected from litter found on South Tarawa, Kiribati and washed using tap water. Wild-type Kiribati eggs were collected in ovitraps, placed in the field for 7 days, and dried over an additional 2–3 days. Eggs were then quantified on the fabric substrate they were laid on and placed into individual containers with food capsules containing the cricket-based diet. As a control, an MRC made of cardboard with a PET lining and dimensions of W 155 mm × H 76 mm × D 60 mm was used. Each container opening was covered with a mesh bag and containers were placed in either 15 cm × 15 cm × 15 cm or 20 cm × 20 cm × 30 cm cages to ensure no wild-type mosquitoes escaped. Cages were positioned around Tungaru Central Hospital in shaded areas with low risk of disruption (Appendix A). Water temperature was measured twice daily for the first three days of larval development using a mercury-in-glass thermometer. Emergence rate was determined seven days post-hatching. Cages were placed in a refrigerator for approximately 3 min to immobilize mosquitoes. Pupae skins, adult mosquitoes, and dead adult mosquitoes inside the container were counted. Emergence rate was calculated as the number of pupae skin as a proportion of the total number of eggs hatched per container. Pupae skins were more intact than adults at time of counting and, therefore, were used as the most reliable measure of emergence rate.

#### 2.8.2. Cost Analysis

A cost analysis was performed to outline the end-to-end process of each proposed waste container release strategy and the assigned costs (in AUD) of each step (Appendix A). As mosquito egg capsules would be prepared at a global office, costs for mosquito larval food, egg production, and importation were not included in this analysis and are considered sunk costs. To conduct the cost analysis, initially, assumptions of release time and resource requirements were established based on the release area in South Tarawa, Kiribati and the World Mosquito Program’s standard operating procedures (Appendix A: Assumptions and inputs). These inputs then informed the cost drivers for the end-to-end process of each release strategy (Appendix A: Cost drivers for releases).

An estimate of the total number of adult mosquitoes released by the different strategies was based off calculations in Appendix A: Mosquito release estimation. It was assumed that a percentage of the community setting up containers appropriately would decrease over subsequent rounds of releases based on observations of field deployments in Queensland. For community-led releases, this was estimated at ~20%, and for staff-led releases, it was estimated to be ~10% per subsequent round. These are estimations based off historical information and to be used for indicative purposes only. Emergence rates were based off results in the waste container field trial conducted in Kiribati.

## 3. Results

### 3.1. Cricket and Black Soldier Fly Meal Offer Viable and Environmentally Friendly Protein Sources for Mosquito Larval Diet

To improve the environmental impact of larval diets, we tested substituting beef liver powder with insect-based protein sources, such as cricket and black solider fly meal. These protein sources offer lower environmental impacts. We first compared the cricket- and fly- based diets with the control diets of beef-based IAEA diet and Tetramin tropical tablets, a commercially available fish food. Based on the success of these diets supporting mosquito rearing, we then tested packing the alternative diet and eggs together inside capsules. In the four-diet comparison experiment, a small but statistically significant increase in emergence rates was observed when using a beef-based diet (pairwise comparison, beef: Tetramin, *p* = 0.0131 *; beef: cricket, *p* = 0.0406 *) (Figure 1a). However, this increase was only observed once, and subsequent experiments indicated no difference between the diet formulations (one-way ANOVA, *p* > 0.05 for all pairwise comparisons) (Figure 2a,b). Emerged females were sampled and measured for wing length and *Wolbachia* density. Mosquitoes reared on a beef-based diet measured statistically longer wing lengths compared to fly and Tetramin diets (pairwise comparison, beef: fly, *p* = 0.0068 **; beef: Tetramin, *p* = 0.0002 ***) (Figure 1b). The difference between these wing length averages is less than 0.15 mm, and all data sets average above 2.75 mm, which is within the expected range of 2.50 and 3.00 mm [51]. This difference was not observed in the follow-up capsule experiments where diet was not found to have a significant effect on wing lengths (Mann–Whitney, beef: fly, *p* = 0.2268; beef: cricket, *p* = 0.7719) (Figure 2c). *Wolbachia* density of the emerged adults was also determined. The insect-based diet reared adults measured comparable (Kruskal-Wallis, chi-squared = 3.131, *p* = 0.3719) (Figure 1c) or significantly higher *Wolbachia* densities compared to beef-based diets, and no drop-out of *Wolbachia* was observed (Mann–Whitney, beef: fly, *p* = 0.0151 *; beef: cricket, *p* < 0.0001 ****) (Figure 2d). Based on the success of the cricket-based diet, the remaining experiments used the cricket-based larval diet, as it was convenient to source at the time of research.

### 3.2. Waste Containers Support Successful Aquatic Development

If egg releases are to be scalable, potential incidental environmental impacts of this method must also be considered. Therefore, we tested reusing waste containers already in existence as MRCs to significantly decrease waste production. We investigated the viability of applying these methods to rear mosquitoes inside four different waste containers. To assess survival from larvae to adulthood in different waste container types, emergence rates were determined. Emergence rates were not significantly different between mosquitoes reared in control containers and a variety of waste containers (*p* > 0.05 for all pairwise comparisons) (Figure 3a and Appendix A). These results demonstrated that waste containers, such as aluminum cans, PET bottles, tin cans, and glass bottles, offer a viable rearing receptacle for *Ae. aegypti* mosquitoes. Emerged females were sampled and measured for wing length and *Wolbachia* density. In multiple repeat experiments, wing lengths remained consistent between mosquitoes reared in different container types (pairwise comparisons, *p* > 0.05), averaging between 2.55 and 2.65 mm (Figure 3b and Appendix A). Similarly, *Wolbachia* density was not significantly affected by rearing in different containers under laboratory conditions (Kruskal–Wallis, *p* = 0.2547) (Figure 3c and Appendix A).

### 3.3. Kiribati Case Study: Waste Containers Tested in the Field Performed Similarly to Control Containers

To assess the viability of waste container egg releases in a real-world scenario, a case study in South Tarawa, Kiribati was conducted to assess the social, political, and economic factors that would influence the success of waste container egg releases.

Preceding experiments demonstrated that using waste containers as MRCs is viable under laboratory conditions. Next, we validated this under field conditions using Kiribati wild-type mosquitoes and measuring the emergence rate and mosquito death rate inside containers to determine the performance of mosquito development. Across the five container types, no significant differences were detected between emergence rates (one-way ANOVA, *p* = 0.45) (Figure 4a). However, we observed very low emergence rates in all groups, most likely caused by field conditions where more factors, such as changing environmental conditions and longer egg collection time, can influence aquatic development or affect egg hatch rate. The average water temperature in each container type ranged between 30.50 °C and 31.63 °C. The average temperature of rearing water across all containers in the field was 31.40 °C. Adult mosquito death was also quantified, with death occurring in all container types at rates between 8–12%. No significant differences were detected across the container types (one-way ANOVA, *p* = 0.71) (Figure 4b).

### 3.4. Cost Analysis Reveals Significant Cost Reductions and Environmental Benefits through Waste Container Release Method

A cost analysis was performed to assess the economic viability of implementing a waste container MRC release method. Based on known availability and suitability of the four alternative MRC sources (see discussion), aluminum cans were considered the most appropriate for use in the context of Kiribati. In addition to waste container type, community engagement strategies were also assessed, as they provide an alternative method for the distribution of MRCs. Churches are a central gathering spot in Kiribati that could be appropriate for mass community engagement and recruitment [52]. Based on these insights, we developed four different strategies for acquiring and distributing MRCs (Table 1 and Appendix A). A cost analysis was performed to compare each strategy. The two main factors compared were the container type (aluminium can or a commercially produced cardboard container) and the method of distribution (community-led or staff-led).

The results of the cost analysis are detailed in Table 2, and the full analysis is provided in Appendix A. The different costs (AUD) were compared for five rounds of releases across four strategies. Strategy 1 was the most cost-effective, at a budgeted deployment cost of AUD 0.14 per person. This was significantly lower than Strategies 2, 3, and 4 which were budgeted to cost AUD 0.68, AUD 2.72, and AUD 4.15 per person, respectively. The biggest contributors to cost difference were the reduced cost required to source and prepare aluminum cans and the increased staff salary associated with staff-led distribution.

## 4. Discussion

The release of *Wolbachia*-infected *Ae. aegypti* as a method of reducing mosquito-borne disease incidence is very promising [15]. Moreover, multiple field trials have demonstrated that it is possible to establish *Wolbachia* in native mosquito populations across large areas [13,14,15,19,20,53]. To apply this method in more locations, effective and sustainable mosquito-rearing and release methods are required to ensure cost efficiency and applicability in different socio-geographical contexts. Egg releases offer an attractive method of mosquito release because they limit onsite resource requirements and can improve public acceptance and community engagement [14,18,19]. We, therefore, investigated how three main components of egg releases (larval diet, egg encapsulation, and MRCs) could be improved to decrease the environmental footprint. We demonstrated that cricket- and fly-based larval diets supported similar emergence rates, wing lengths, and *Wolbachia* density to a beef-based larval diet, including when encapsulated with eggs inside of water-soluble capsules. We further showed that waste containers (aluminum and tin cans, PET, and glass bottles) can be used as environmentally friendly mosquito release containers with minimal impact on container success, and we identified cost-effective strategies for deployment.

Using insects as an alternative source of protein offers an opportunity to improve the sustainability of larval diet [22,23,24,25,54]. Promisingly, cricket and black soldier fly meal provide similar nutritional values as beef [25,26], and we showed that they provide a sufficiently nutritionally rich larval diet for survival and healthy adults, which is important to ensure competitiveness with wild-type populations. Promisingly high *Wolbachia* densities, with no *Wolbachia* drop-out, were observed across all diet, capsule, and waste container experiments. This is an important measure, as maintaining high *Wolbachia* frequencies is important for successful introgression and persistence in the environment [55,56]. To implement these strategies at scale, further investigations into the availability and quality assurance of insect meal are required. Promisingly, unlike beef, rearing and slaughtering insects was not illegal in any countries at time of publication [57,58]. Furthermore, the encapsulation method was beneficial, as it increased the ease of on-site distribution in egg releases and reduced potential waste production by replacing the plastic bags that were previously used. While initially promising, further testing is required under longer and more varied storage conditions.

One of the potential drawbacks of egg releases is the waste that is produced by placing single-use MRCs in the field. Due to the inadvertent adverse health effects of waste, such as increased risk of hygiene-related diseases [31,32,33,34,37], and existing waste problems in release areas, such as Kiribati [37,59,60], it is important to consider the environmental impact of *Wolbachia* mosquito releases. Our study demonstrated that waste containers make suitable MRCs for egg releases that would have a net negative environmental footprint. To assess the applicability of waste container egg releases, we looked to the Pacific island nation of Kiribati as a case study. Kiribati has been affected by *Aedes*-borne disease outbreaks, such as dengue and chikungunya virus, infecting approximately 12,000 people, or 21% of the population, in 2015 [61]. Kiribati would particularly benefit from environmentally-friendly release methods because it is geographically isolated, scarce in natural resources, and has a limited ability to process waste items or recycle locally, to the extent that the majority of waste is shipped for offshore processing [36,37]. Waste management investigations specific to Kiribati have highlighted that solid waste and pollution can affect coastal fisheries, which are already threatened by overfishing, mining, and coastal erosion [59]. In 2014, only 38% of the total waste in South Tarawa was collected and disposed of by council authorities; 26% was disposed of on-site by communities, 1% was recycled, and 35% was illegally dumped on beaches or in the ocean [60]. As such, it is important to consider the environmental and potential social impacts of implementing widespread egg releases in settings such as this.

After demonstrating the viability of waste containers in controlled conditions, we tested the same container types in field conditions in South Tarawa, Kiribati with wild-type mosquitoes to understand the feasibility of this concept. Due to logistical restraints, only field-collected eggs were available for the on-site experiment. In ovitraps, eggs are laid on a fabric substrate, which makes brushing for encapsulation difficult and was why eggs remained on the substrate for this field trial. Emergence rates reduced significantly in the field compared to laboratory experiments, most likely due to a range of external and variable environmental conditions. Eggs were collected in ovitraps over a 7-day period prior to drying down. An extended collection period could result in pre-hatching, so low hatch rates may have been a contributing factor. In addition, environmental factors observed included high average water temperature inside the containers (31.4 °C), physical disturbances, and sun and rain exposure. In the case of zero emergence, because no adult death was observed in these containers, failed emergence was most likely due to low hatch rates resulting in the over-provision of food, which increases the risk of water fouling. However, reduced emergence rates in the field remained consistent across all container types, and no statistically significant differences were detected between the average emergence rates of different containers used in the field. Promisingly, in both laboratory and field settings, waste container emergence rates were comparable with respective controls. It is evident that further field testing must be conducted to assess how field emergence can be optimized and to ensure release levels are as predicted. Additional considerations need to be made and tested for *Wolbachia*-infected mosquitoes whose survival and *Wolbachia* densities under varied temperature and nutritional environments may be affected differently. For example, it was observed that rearing at high temperatures resulted in lower larval survival [62]. In addition, high rearing temperatures were linked to a decrease in *Wolbachia* density when cycling between 26–36 °C; therefore, ensuring MRCs are placed in an appropriate setting where temperatures can remain low is crucial [55]. To address the issue of disturbances, it is important to ensure adequate community consultation and engagement prior to implementation. The Public Acceptance Model has been developed specifically for the citywide deployment of *Wolbachia* mosquito releases to help guide what needs to be taken into consideration [20]. For example, specific to Kiribati, there exists a collectivist society where decision making regarding community development projects occurs in church groups or meeting houses called mwaneabas [52]. Understanding and aligning with this collectivist structure will boost community engagement, ensuring the success of egg releases.

After validating the four waste container types as viable MRC options, we then conducted a literature analysis to assess which container would be most suitable in the context of Kiribati. One of the most significant threats to the waste container MRC concept is if the container has already been repurposed in the community. In the context of Kiribati, PET and glass bottles are repurposed for storage and solar water disinfection due to a health campaign led by the Kiribati government [63,64]. The existing value of these containers means they would be problematic as MRCs due to an increased risk of interference and conflict with existing governmental health advice. Alternatively, there remains tin and aluminum cans. Tin cans are suboptimal for myriad reasons. A weakness for tin cans is that most are found in landfills, meaning acquiring these containers would work against the Kiribati Ministry of Environment, Lands, and Agriculture Development (MELAD) waste management operations that aim to remove waste from the environment. In addition, they could be dangerous to work with due to sharp edges and would require additional cleaning, depending on the initial contents. Tin cans are also more likely to provide a breeding site for native mosquitoes due to wide openings and rough edges [65]. Promisingly, aluminum cans offer a favorable MRC option, as they are readily available at the MELAD recycling facility. This is due to the recycling initiative in South Tarawa where community members receive AUD 0.05 for recycling containers [35]. For aluminum cans to work as MRCs, coordination and partnership with MELAD is required to ensure non-damaged containers are available for collection, and adequate community engagement is required to ensure cans are not discarded prior to mosquito release completion.

While waste containers offer a great opportunity to reduce the environmental cost of egg releases, it is important for scalability that this method is cost-efficient. To assess the cost-effectiveness of the waste container egg release method, a cost analysis was conducted to compare the end-to-end processes between utilizing aluminum cans or cardboard MRCs and implementing community- or staff-led distribution methods. This analysis highlights that utilizing waste containers as MRCs significantly decreased costs by at least 34% while minimizing the environmental footprint, regardless of the distribution method. This was primarily due to a lower purchasing and importation cost. It was evident that community-led distribution combined with aluminium cans (strategy 1) is the most financially viable option (AUD 0.14 per person). Strategy 1 offers a scalable egg release with minimal environmental impact. The main risk management required for strategy 1 is ensuring adequate and high-quality community participation. Alternatively, utilizing aluminum cans also significantly reduced costs in the staff-led distribution method, strategy 3 (AUD 2.72 per person). When considering which method is optimal, a cost-benefit analysis is required to take into account that distributing directly, i.e., staff-led, is likely to result in higher compliance and mosquito release rates, resulting in the release of approximately 34% more mosquitoes, as projected in the cost analysis (Appendix A). Regardless of the distribution method employed, utilizing waste containers in Kiribati as MRCs would significantly reduce financial costs, as well as environmental impacts.

In conclusion, we provided evidence for an environmentally conscious release method. Utilizing larval diets and mosquito release containers with a reduced environmental footprint can also reduce potential negative social impacts and financial costs while not compromising the product quality. This study supports contributing to a greener economy as an economically and socially beneficial option. Each release location must conduct a similar assessment to determine which reusable container is most appropriate within its unique context. This method will aid the scalability of the *Wolbachia* introgression method in new global projects while reducing both logistical costs and potential spillover environmental impacts to, ultimately, reduce the global burden of mosquito-borne diseases.

## Figures and Tables

**Figure 1 pathogens-11-00373-f001:**
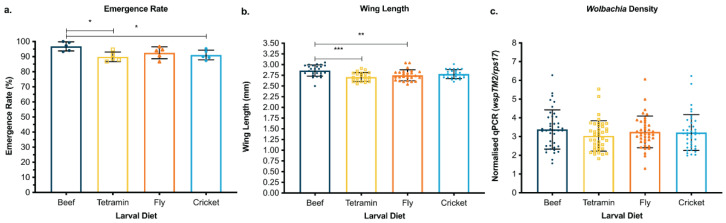
Development and adult fitness comparison between *w*Mel-infected *Ae. aegypti* mosquitoes fed on four different larval diets. Aus *w*Mel *Ae. aegypti* eggs were hatched and reared using different larval diets. Each cohort consisted of 5 populations of 150 larvae. (**a**) Emergence rate, (**b**) wing length, and (**c**) *Wolbachia* densities are shown. Each data point represents one cup (emergence), one wing from an individual mosquito (wing length), or one mosquito (*Wolbachia* density). An ANOVA (*p* < 0.05 *, *p* < 0.01 **, *p* < 0.001 ***) was performed on the dataset, followed by Tukey’s multiple comparison test. Error bars represent standard error of the mean.

**Figure 2 pathogens-11-00373-f002:**
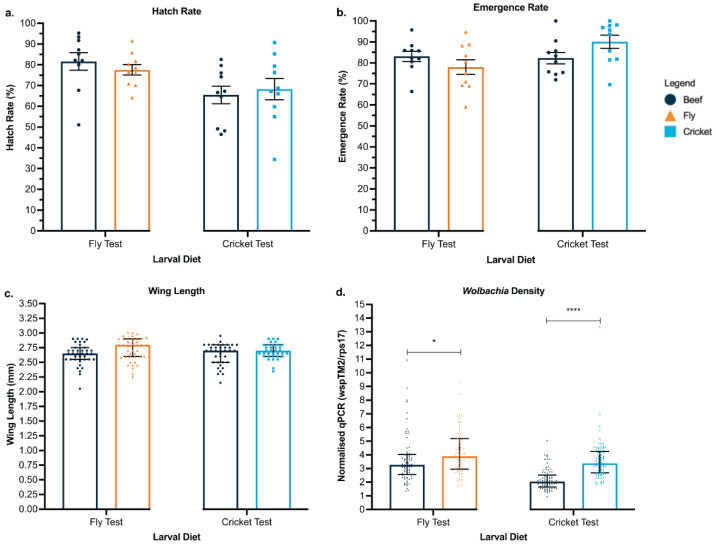
Development and adult fitness comparison between *w*Mel-infected *Ae. aegypti* packaged in water-soluble capsules with different diets. Aus *w*Mel *Ae. aegypti* eggs and beef-, fly- or cricket-based larval diets were packaged into water-soluble capsules before hatching. Two experiments are represented in each graph. The first experiment compared beef liver powder to black soldier fly meal, and the second experiment compared beef liver powder to cricket meal. (**a**) Hatch rate, (**b**) emergence rate, (**c**) wing length, and (**d**) *Wolbachia* densities are shown. Each data point represents one cup (hatch and emergence), one wing from an individual mosquito (wing length), or one mosquito (*Wolbachia* density). Hatch and emergence rate data were analyzed by ANOVA, followed by Tukey’s multiple comparison test, and are shown as the mean and standard error. Wing length and *Wolbachia* density data were analyzed by Mann–Whitney test, and data are shown as medians with interquartile ranges. *p* = 0.0151 *; beef: cricket, *p* < 0.0001 ****.

**Figure 3 pathogens-11-00373-f003:**
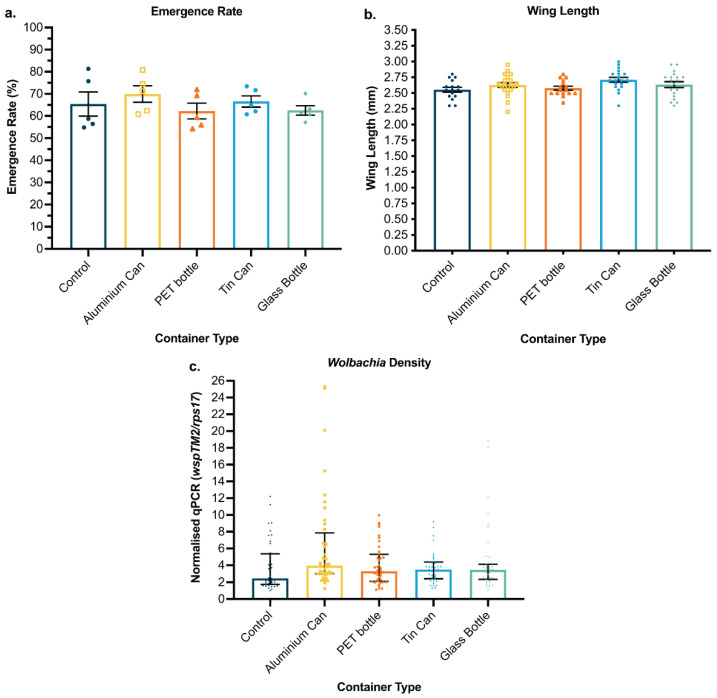
Development and adult fitness comparison between *wMel*-infected *Ae. aegypti* mosquitoes reared in five different containers. Aus *w*Mel *Ae. aegypti* eggs were fed a cricket-based diet inside capsules. This is representative of two independent repeat experiments and data for the other repeat is presented in Appendix A. Each cohort consisted of five populations of 75–100 eggs. (**a**) Emergence rate, (**b**) wing length, and (**c**) *Wolbachia* densities are shown. Each data point represents one cup (emergence), one wing from an individual mosquito (wing length), or one mosquito (*Wolbachia* density). Emergence rate and wing length data were analyzed by ANOVA, followed by Tukey’s pairwise comparison test, and are shown as the mean and standard error. *Wolbachia* density data were analyzed by Kruskal–Wallis test and are shown as medians with interquartile ranges.

**Figure 4 pathogens-11-00373-f004:**
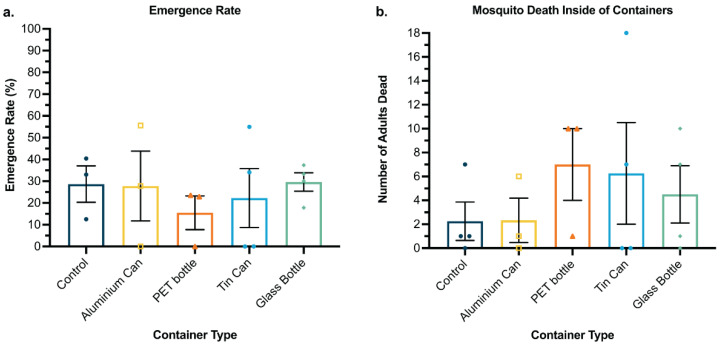
Emergence rate and mosquito death inside container. Comparison between wild-type (South Tarawa) mosquitoes reared in four different containers in field conditions. Wild-type *Ae. aegypti* Kiribati eggs were hatched and fed cricket-based diet inside capsules. Each cohort consisted of four populations of 75–100 eggs. (**a**) Emergence rate and (**b**) mosquito death inside containers are shown. Data are shown as the mean and standard error. ANOVA was performed on the dataset. No significant differences were detected. Error bars represent standard error of the mean.

**Table 1 pathogens-11-00373-t001:** Outline of four egg release strategies.

Strategy	Container	Source	Distribution Method
1	Aluminum can	Recycling facility	Community
2	Cardboard MRC	Detpak, imported into Kiribati	Community
3	Aluminum can	Recycling facility	Staff
4	Cardboard MRC	Detpak, imported into Kiribati	Staff

**Table 2 pathogens-11-00373-t002:** Comparison of the four release strategies. Costs given in AUD.

	Strategy 1	Strategy 2	Strategy 3	Strategy 4
Method of distributing container	Community distribution	Community distribution	Staff-led distribution	Staff-led distribution
Materials used	Aluminum can	Cardboard + PET lining	Aluminium can	Cardboard + PET lining
Mosquito release rounds	5	5	5	5
Duration	10-week deployment	10-week deployment	10-week deployment	10-week deployment
Projected total number of mosquitoes released over 5 release rounds	807,429	807,429	1,229,513	1,229,513
Total cost of sourcing and preparing containers	$568.80	$21,684.00	$568.80	$21,684.00
Total staff salary cost	$5078.00	$7200.00	$120,758.00	$168,000.00
Total deployment cost	$7896	$38,387	$153,150	$233,768
Deployment cost per person	$0.14	$0.68	$2.72	$4.15
Supplier availability	Collect from recycling center	Collect from communities	Collect from recycling center	Import from overseas
Recyclable in Kiribati	Yes	No	Yes	No
Reusable	Yes	No	Yes	No
Require addition of labels	Yes	No	Yes	No
Environmental impact	Neutral– positive	Negative	Neutral– positive	Negative
Local supplier	Yes	No	Yes	No
Mosquito release quality	Low	Low	Medium	Medium

## Data Availability

All data are provided within the text and Appendix A.

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
