# Peer review of "Trash to Treasure: How Insect Protein and Waste Containers Can Improve the Environmental Footprint of Mosquito Egg Releases"

_pathogens, 2022, doi:10.3390/pathogens11030373_

Round 1

Reviewer 1 Report

This is well written manuscript with very cool experimental setup. Authors 1. compare insect-based diets to cut the cost on previous beef-based diet for larvae rearing.

2. test encapsulation strategy to ease transportation, on-site distribution, and reduce waste generation.

3.  compare waste containers to release mosquitoes to reduce environmental hazards.

They do so by also tracking development, fitness, and Wolbachia density to ensure persistence of released Wolbachia in released healthy mosquitoes. It is all done meticulously and I congratulate the authors for designing and executing this study. 

However, below are a few things to be addressed by the authors:

1. L61 - Please explain what unwanted experimental outcomes authors are referring to. While it is clear later in the discussion section, it would be good to enlist it ahead in the introduction to help readers understand the call of duty. 

2. Provide the individual data point values in Figure 4 to provide a better visual of data distribution, just as done in other figures.

3. Some information in discussion is redundant with that provided in the intro section. Revising and readjustment will help distinguishing the two as already mentioned in point 1 above.

Author Response

Dear Reviewer 1,

Thank you for your comments. We have made every effort to address them all. Our edits have been tracked changed and responses to comments are provided below.

L61 - Please explain what unwanted experimental outcomes authors are referring to. While it is clear later in the discussion section, it would be good to enlist it ahead in the introduction to help readers understand the call of duty.

Added example “such as waste accumulation due to single-use MRCs.” to L63

Provide the individual data point values in Figure 4 to provide a better visual of data distribution, just as done in other figures:

Edited figure 4 to include individual data points. In addition, we also added discussion on zero emergence cases in the discussion (L431-433).

Some information in discussion is redundant with that provided in the intro section. Revising and readjustment will help distinguishing the two as already mentioned in point 1 above:

Removed unnecessary points already covered in the introduction (L360 and L376).

Reviewer 2 Report

The authors present a fascinating, well-structured study that covers a range of aspects, from the laboratory to the field.  I have annotated this on the manuscript, but there are two key issues I feel I must highlight.

Although the statistics are appropriately chosen, the reporting of the statistics is sub-par. Exact p-values, not approximations, are required. Other statistical indicators must be given as p-values alone are not good enough to judge statistical validity.

Furthermore, despite the fact that the discussion is very clearly and logically written, everything in it is covered before in the manuscript, so technically it is a little on the descriptive side. Some additional points and insights would bring the discussion to the next level.

Author Response

Dear Reviewer 2,

Thank you for your comments. We have made every effort to address them all. Our edits have been tracked changed and responses to comments are provided below.

L48- It may be an idea to start this as a new paragraph.: New paragraph started.

L61- Give some examples please.: Added example “such as waste accumulation due to single-use MRCs.”to L63

L110- Please give the preparation in brief- at least the components.: Added list of individual components in L116-117 “beef-liver powder, tuna meal and brewer’s yeast”

L122- counted or weighed? Please specify.: Clarified quantification method as “by counting” in L130.

L165- Please provide the references for the statistical tests used.: References for statistical tests used added to section 2.7, including the addition of Mann-Whitney in L183. Remaining reference numbers adjusted.

L173- A map highlighting the field trial site would be good, even if included as supplementary material.: Map added to supplementary material, Figure S1.

L220- Please give the exact value rather than an approximation.: Converted all in text p-values to provide exact values.

L225/231/257/292- Please quote the F statistic and degrees of freedom as well./ please quote the chi-squared statistic as well./ Again, statistics reporting to be tightened up please./ other statistical indicators.: Degrees of freedom, F-statistic and chi-squared statistic for all tests were added to supplementary information in the form of a table- Supplementary table 4. In some cases it was not reader friendly to include all these in text, in particular when referencing multiple tests at once.

L306- There were no significant difference between the groups.: Mention of “tukey’s multiple comparison” removed from figure 4 caption.

Furthermore, despite the fact that the discussion is very clearly and logically written, everything in it is covered before in the manuscript, so technically it is a little on the descriptive side. Some additional points and insights would bring the discussion to the next level:

Removed unnecessary points already covered in the introduction (L360 and L376). The remainder of the discussion covers additional points on the specifics of waste disposal issues in Kiribati, discussion of the field trial and factors that could impact application of this research in the field at scale.

Reviewer 3 Report

Well described paper for and valuable environmental friendly protocol proposal. It will be nice to support graphical abstracts and/or infographics to reach a broader range of readers from multidisciplinary fields.

Future remarks would improve the quality of the paper. If a general abstract of future remarks targeting policymakers, investors and funding agencies highlighting the potential contribution of the approach to a greener economy and societal benefits will be impactful.

Author Response

Dear reviewer 3,

Thank you for your comments. We have made every effort to address them all. Our edits have been tracked changed and responses to comments are provided below.

Support graphical abstracts and/or infographics to reach a broader range of readers from multidisciplinary fields:

Added supporting graphical abstract.

Future remarks would improve the quality of the paper. If a general abstract of future remarks targeting policymakers, investors and funding agencies highlighting the potential contribution of the approach to a greener economy and societal benefits will be impactful:

Future remarks added to the final discussion paragraph to emphasise the economic and social benefits of a greener economy (L495-498).